# Charge Trapping and Emission Properties in CAAC-IGZO Transistor: A First-Principles Calculations

**DOI:** 10.3390/ma16062282

**Published:** 2023-03-12

**Authors:** Ziqi Wang, Nianduan Lu, Jiawei Wang, Di Geng, Lingfei Wang, Guanhua Yang

**Affiliations:** 1State Key Lab of Fabrication Technologies for Integrated Circuits, Institute of Microelectronics, Chinese Academy of Sciences, Beijing 100029, China; wangziqi@ime.ac.cn (Z.W.);; 2Laboratory of Microelectronic Devices and Integrated Technology, Institute of Microelectronics, Chinese Academy of Sciences, Beijing 100029, China; 3University of Chinese Academy of Sciences, Beijing 100029, China

**Keywords:** carrier capture and emission, c-axis aligned crystalline indium-gallium-zinc-oxide (CAAC-IGZO), first principles calculation, transistor instability effect

## Abstract

The c-axis aligned crystalline indium-gallium-zinc-oxide field-effect transistor (CAAC-IGZO FET), exhibiting an extremely low off-state leakage current (~10^−22^ A/μm), has promised to be an ideal candidate for Dynamic Random Access Memory (DRAM) applications. However, the instabilities leaded by the drift of the threshold voltage in various stress seriously affect the device application. To better develop high performance CAAC-IGZO FET for DRAM applications, it’s essential to uncover the deep physical process of charge transport mechanism in CAAC-IGZO FET. In this work, by combining the first-principles calculations and nonradiative multiphonon theory, the charge trapping and emission properties in CAAC-IGZO FET have been systematically investigated. It is found that under positive bias stress, hydrogen interstitial in Al_2_O_3_ gate dielectric is probable effective electron trap center, which has the transition level (ε (+1/−1) = 0.52 eV) above Fermi level. But it has a high capture barrier about 1.4 eV and low capture rate. Under negative bias stress, oxygen vacancy in Al_2_O_3_ gate dielectric and CAAC-IGZO active layer are probable effective electron emission centers whose transition level ε (+2/0) distributed at −0.73~−0.98 eV and 0.69 eV below Fermi level. They have a relatively low emission barrier of about 0.5 eV and 0.25 eV and high emission rate. To overcome the instability in CAAC-IGZO FET, some approaches can be taken to control the hydrogen concentration in Al_2_O_3_ dielectric layer and the concentration of the oxygen vacancy. This work can help to understand the mechanisms of instability of CAAC-IGZO transistor caused by the charge capture/emission process.

## 1. Introduction

Benefited from the high mobility and good large-area uniformity, indium-gallium-zinc-oxide (IGZO) has received a lot of attention since it was discovered in 2004 [1]. After years of development, this material is making its entry into the display industry, thanks to its better performance than amorphous silicon [2,3,4,5]. Recently, a new crystalline oxide semiconductor called c-axis aligned crystalline IGZO (CAAC-IGZO) has also caught the attention of researchers. Field-effect transistors (FETs) with the CAAC-IGZO channel layer exhibit an extremely low off-state leakage current (~10^−22^ A/μm) [6,7], which helps to further reduce the leakage power in-memory application. More importantly, the mobility of the CAAC-IGZO FET doesn’t degrade at high temperatures [8]. In terms of the potential advantages of extremely low power, high mobility, capacitorless architecture and back-end-of-line (BEOL) compatibility [9,10,11], the CAAC-IGZO FET has promised to be an ideal candidate for DRAM applications [7,12,13].

Despite its benefits, the instability of CAAC-IGZO FETs greatly limits their application. These instabilities are mainly reflected in the drift of the threshold voltage when the device works continuously in various stress, which are usually distinguished: negative bias stress (NBS) and positive-bias stress (PBS). That is, NBS instabilities shift the transfer curve of the FET negatively, while PBS shifts it positively. To clearly explain the fundamental performance of IGZO transistors, a great deal of possible mechanisms have been proposed [14,15,16,17,18,19], for instance, charge trapping processes at the interfaces and/or in the dielectric, the creation and impact of deep traps in active layer, the absorption of oxygen or water molecules at the channel interface, the removal of oxygen interstitials, the capture of electrons by oxygen vacancies, the reduction of peroxide concentration, and the desorption and diffusion of hydrogen, and so on. However, due to the complexity of the dielectric layer and semiconductor layer, the origin of the bias instabilities in CAAC-IGZO FETs is still controversial. On the other hand, thanks to the sensitivity of the transistors to various factors, such as the deposition conditions, annealing conditions and gate-dielectric material, it is difficult to isolate the impact of different origins on reliability. Therefore, to better develop high-performance CAAC-IGZO FETs for DRAM applications, it’s essential to uncover the deep physical process of charge transport mechanisms in CAAC-IGZO FETs.

In this work, we systematically investigate the charge trapping and emission properties in CAAC-IGZO FETs based on a first-principles calculations. Then, by combining the nonradiative multiphonon theory, the trapping process properties of several intrinsic defects in CAAC-IGZO stack with Al_2_O_3_ dielectric have been discussed in detail. Finally, the influence of charge trapping process on the reliability of devices has also been discussed under positive bias stress (PBS) and negative bias stress (NBS).

## 2. Computation Methodology

To uncover the charge trapping and emission properties, a general FET is selected as a prototype device. Figure 1 shows the illustration of the carrier trapping process in CAAC-IGZO FET with active layer of CAAC-IGZO and gate dielectric of Al_2_O_3_. The first-principles calculations were utilized based on the density functional theory (DFT) of the first-principle plane-wave pseudopotential method. All calculations were performed by the Vienna Ab Initio Simulation Package (VASP) software [20]. The projector augmented wave (PAW) [21] pseudopotential was used to describe the interactions between the nucleus and valence electrons. The generalized gradient approximation (GGA) with the Perdew-Burke-Ernzerhof (PBE) functional was selected to model the exchange and correlation interactions [22]. The cut-off energy for the wave function was set to 500 eV. All structural models were fully relaxed using the conjugate gradient algorithm until the atomic force was less than 0.01 eV/Å.

The hybrid functional of Heyd-Scuseria-Ernzerhof (HSE06) was used to ensure the accuracy of bandgap calculation and avoid the small bandgap under GGA-PBE function [23]. 30% and 15% of PBE were replaced with Hartree-Fock function in the calculation of Al_2_O_3_ and IGZO, respectively. To avoid overestimation of metal *d* bands, the GGA+*U* method was used to include an on-site Coulomb correlation interaction between the localized metal *d* electrons [24]. The parameters of *U* = 8.0 eV, 8.0 eV, 8.0 eV and under PBE function and *U* = 3.5 eV, 4.0 eV, 4.0 eV under HSE06 function were selected for In 3*d*, Ga 3*d* and Zn 3*d* orbitals, respectively.

Figure 2 shows the crystal model of α-Al_2_O_3_ and CAAC-IGZO. Al_2_O_3_ as a gate dielectric is often grown by ALD to form an amorphous structure. However, Choi et al. [25] confirmed that the features of native defect in Al_2_O_3_ are not strongly dependent on the phase, and the results in the α-Al_2_O_3_ are representative and applicable to an amorphous structure. The optimized lattice parameters for the perfect α-Al_2_O_3_ crystal are *a* = *b* = 4.767 Å and *c* = 13.028 Å, which are good agreement with the experimental values with *a* = *b* = 4.656 Å and *c* = 13.140 Å [26]. To simulate the defects, a supercell of 2 × 2 × 1 with 120 atoms was used. The optimized lattice parameters for the perfect CAAC-IGZO crystal are *a* = *b* = 3.345 Å and *c* = 26.083 Å, which are good agreement with the experimental values with *a* = *b* = 3.2948 Å [27] and *c* = 26.071 Å [28]. To reduce the interaction of defects in the *a-b* plane, a supercell with 112 atoms was used. The supercell was obtained by setting lattice vectors to (420), (040), and (221) for the IGZO crystal and reducing the lattice constant *c* to one-third [29]. To have an insight into the capture process of the CAAC-IGZO FETs with Al_2_O_3_ dielectric, a variety of different intrinsic defects are considered, which may widely exist and have been suggested as the cause of reliability issues [30,31,32,33,34,35] in oxide semiconductor materials, for instance, oxygen vacancy (V_O_), oxygen interstitial (O_i_), hydrogen interstitial (H_i_), hydrogen substituted oxygen (H_O_) and hydroxyl interstitial ((OH)_i_), and so on. Here, the defect of Ho can be regarded as a coexistence of H_i_ and V_O_. And the defect of (OH)_i_ can be regarded as a coexistence of H_i_ and O_i_. In IGZO crystalline, because of the different chemical environment from nearby metal atoms, four kinds of oxygen vacancy will be created by removing the selected oxygen atoms, as well as two kinds of oxygen interstitial, as shown in Figure 2b.

## 3. Result and Discussion

### 3.1. Formation Energy and Transition Level

As mentioned above, the CAAC-IGZO FETs may be affected by a variety of defects. To identify which defects have an impact on the FET, the formation energy of all kinds of defects will be discussed firstly. The formation energy of defect can be obtained as [37],
(1)$$E_{for}^{q}=E_{defect}^{q}-E_{perfect}-\sum_{i}n_{i}\mu_{i}+q\left(E_{f}+E_{v}+\Delta V\right),$$
where $E_{defect}^{q}$ represents the total energy of a supercell with the defect in the cell, $E_{perfect}$ is the total energy for a perfect supercell, $n_{i}$ is the number of atoms removed from or added to the supercell to form a defect, $\mu_{i}$ is the chemical potentials of oxygen and hydrogen atom/ion, q represents the number of electrons transferred from or to electron reservoirs [38,39], $E_{f}$ refers to the Fermi level relative to the valence band maximum energy (VBM, $E_{v}$) position in a perfect supercell. The correction term, ΔV, is referred to as the electrostatic potential that is far from the defect in the supercell with respect to the perfect supercell of the same size. Otherwise, in order to determine the position of the defect level in the band gap, one can calculate the transition level. Based on Equation (1), the position of the Fermi level at which the formation energy of charge states *q* equals to the formation energy of charge states *q’* was defined as transition level ε(*q/q’*) [37].

Generally, by using the charged formation energy and transition level, one can clearly understand the stability of defect and the relative position of defect level in the bandgap from an energy perspective. Figure 3 shows the formation energy of each defect under O-rich conditions In this work, the Fermi level of IGZO channel is set at 0.2 eV below $E_{C}$ in the absence of an electric field, which is a reasonable assumption [2,40]. From in Figure 3a, as compared with the other oxygen vacancies in IGZO, Vo_1,_ being nearby an In atom and Zn atom, exhibits the lowest formation energy. This result indicates that Vo_1_ may exist more stably in IGZO than other oxygen vacancies, which is attributed to the weak bonding between In atom and O atom. It is noting that unlike the role of oxygen vacancies in a-IGZO and some transition metal oxides [31,41], oxygen holes do not act as n-type electron donors in CAAC-IGZO. In contrast, the formation energy of interstitial oxygen does not depend significantly on the metal ions in its vicinity. More importantly, our calculations also show the most stable charge states for each intrinsic defect in IGZO and Al_2_O_3_ without an electric field. In the absence of electric field, the Fermi level is 0.2 eV below $E_{C}$ and the most stable charge states for intrinsic defects (V_O_, O_i_, H_i_, H_O_ and (OH)_i_) in IGZO, are 0, −2, +1, +1 and −1, respectively. And the most stable charge states for intrinsic defects (V_O_, O_i_, H_i_, H_O,_ and (OH)_i_) in Al_2_O_3_, are also 0, −2, +1, +1 and −1, respectively. These charged defects will form a charged center in oxide and lead to the fixed oxide charge, and then significantly influence the electrical characteristics of transistor. As a result, H_i_ and H_O_ defects will lead to positive fixed oxide charge, while O_i_ and (OH)_i_ will result in a negative fixed oxide charge.

Figure 4 shows the band alignment and transition level of defects in Al_2_O_3_/IGZO system. In general, the transition levels reveal which defects undergo electron capture or emission. With the position of the Fermi level changes in the presence of electric field, the defects will capture electrons when the Fermi level is higher than the transition level as well as emission electrons when the Fermi level is lower than the transition level. Therefore, if the transition level is closer to the Fermi level, the defect is a more efficient center for electron capture or emission under the electric field. In Figure 4, one can see that the transition level ε (+1/−1) of H_i_ in Al_2_O_3_ is 0.52 eV above the Fermi level of CAAC-IGZO which is closer to the Fermi level, as compared with the transition level ε (+1/−1) of H_O_ in Al_2_O_3_ which is 2.14 eV above Fermi level. This result suggests that the defect H_i_ in Al_2_O_3_ gate dielectric is probable effective electron trap center for CAAC-IGZO channel under PBS. The reason is that when a positive bias voltage is applied, the bandgap of the gate dielectric is shifted downwards, thus allowing the process of electron capture more likely to occur in H_i_ defects in Al_2_O_3_ gate dielectric. The detailed schematics for electron trapping process under PBS can be seen in Figure 5a. In contrary, the transition levels ε (+2/0) of Vo in CAAC-IGZO and ε (+2/0) of Vo in Al_2_O_3_ are distributed at −0.73~−0.98 eV and 0.69 eV below the Fermi level of CAAC-IGZO, respectively. The transition level ε (+2/0) for Vo in CAAC-IGZO and Al_2_O_3_ is closer to the Fermi level, as compared with other defects. The results indicate that the oxygen vacancies in CAAC-IGZO active layer and Al_2_O_3_ gate dielectric are likely to be the electron emission center under NBS, due to the bandgap of gate dielectric shifting upwards thus allowing the process of electron emission. Figure 5b shows the schematics of electron emission process under NBS in detailed. The properties of electron capture or release in these defects will be discussed in more detail in the following.

### 3.2. Electronic Properties of Charge Capture/Emission Process

According to the analysis of the transition levels above, one can determine that H_i_ in Al_2_O_3_ is the possible electron capture center and V_O_ in both Al_2_O_3_ and CAAC-IGZO is the possible electron emission center. To clearly understand the electronic properties of Al_2_O_3_ and CAAC-IGZO with different defects and charge states during the charge transition process, we calculated the density of states (DOS) of Al_2_O_3_ and CAAC-IGZO with different defects and charge states, respectively.

Figure 6 shows the DOS and the local charge density of hydrogen interstitial in Al_2_O_3_ dielectric layer. It is found in Figure 6a that, when the charge state in Al_2_O_3_ with the defects is +1, the protonated hydrogen forms a hydroxide ion with the oxygen ion, at which there is no sub-state in the bandgap. When $H_{i}^{+}$ captures an electron and forms $H_{i}^{0}$, the hydrogen-oxygen bond will be broken and the hydrogen atom moves to the middle of the aluminum ion, as shown in Figure 6b. In this case, a new sub-state DOS arises and is distributed at 3.2 eV above VBM, which is composed of H orbitals and O orbitals. When $H_{i}^{0}$ captures an electron and forms $H_{i}^{-}$, the aluminum ion moves closer to the H atom. The sub-state DOS distributed at 2.6 eV above VBM, composed of H orbitals and O orbitals, which is 0.6 eV lower than $H_{i}^{0}$.

Figure 7 shows the DOS and the local charge density of oxygen vacancies in Al_2_O_3_ dielectric layer. One can see that, when the defects are in the charge state of 0, the electrons are mainly present in the oxygen vacancies. The sub-state DOS distributes at 2.7 eV above VBM, which is composed of Al orbitals and O orbitals. When $V_{O}^{0}$ emits an electron and forms $V_{O}^{+}$, the aluminium ion moves slightly outward from the oxygen vacancy. The sub-state DOS distributes at 4.1 eV above VBM, which is 0.9 eV higher than $V_{O}^{0}$, as shown in Figure 7b. On the other hand, the electron emission process for the oxygen vacancy from $V_{O}^{+}$ to $V_{O}^{2+}$ is very similar to that from $V_{O}^{0}$ to $V_{O}^{+}$ process, again with the aluminum ion moves slightly outward from the oxygen vacancy. The sub-state DOS distributes at 5.4 eV above VBM, which is 1.1 eV higher than $V_{O}^{+}$.

Figure 8 shows the DOS and the local charge density of oxygen vacancies in the CAAC-IGZO active layer. As can be seen in Figure 8 that the electronic properties of four kinds of oxygen vacancies are similar. When the defects are in the charge state of 0, the electrons are mainly present in the oxygen vacancies. The sub-state DOS distributes at 1.6 eV above VBM. For the oxygen vacancies from $V_{O1}$ and $V_{O2}$, the sub-state DOS is mainly composed of In orbitals and O orbitals, while for $V_{O3}$ and $V_{O4}$, the sub-state DOS is composed of both Ga orbitals, Zn orbitals and O orbitals. When $V_{O}^{0}$ emits an electron and forms $V_{O}^{+}$, the metal ions move outward from the oxygen vacancy. The sub-state DOS distributes at about 2 eV above VBM, which is 0.4 eV higher than $V_{O}^{0}$. Similarly, the electron emission process for the oxygen vacancies from $V_{O}^{+}$ to $V_{O}^{2+}$ also arises with the outward movement of metal ions. The sub-state DOS is distributed near the Conduction Band Minimum (CBM).

### 3.3. Kinetics of Charge Capture/Emission Process

To uncover the kinetics of the charge capture/emission process in CAAC-IGZO FETs, it is crucial to know the activation barrier of the transition process. However, the transition level mentioned above only considers the equilibrium energy relationship between the two-defect charge states and ignores the deformation of the defect site when the charge state is changing. Physically, changes in defect configuration and electron-phonon coupling alter the activation barrier of the charge capture/emission process and will significantly influence the kinetics of the charge capture/emission process. In order to better understand the kinetics of the charge capture/emission process, it is essential to further consider the relationship between the defect formation energy and configuration. The theoretical model used in this work is called as nonradiative multiphonon theory (NMP) [42,43], which has been proven to correctly describe the charge trapping of oxide defects in transistors [44].

#### 3.3.1. Nonradiative Multiphonon Transition Process

Generally, if a defect is neutral in state $V_{1}$ and negatively charged in state $V_{2}$, the atomic equilibrium configuration is different in each state. Here, the different states are denoted by $e_{1}$ and $e_{2}$. Since the real motion of atoms is highly complex, one usually uses a single reaction coordinate e to hold all 3N coordinates of the N atoms under consideration. The total energy consists of contributions from the ionic system, the electronic system, and a coupling term in each state. The coupling term describes the shift in the equilibrium positions and the change of the vibrational frequencies. In most case, the energy-coordinate relationship can be approximated as a quadratic function for small displacement. This simplification can reasonably reflect the essence of the capture process [45]. Another simplification is first-order approximation [45,46], which assumes that only one vibrational mode contributes to the electronic transition process. Based on these approximations, the relationship of total energy to reaction coordinates is usually written as [44],
(2)$$E_{V_{i}}=\frac{1}{2}M\omega_{i}^{2}{\left(e-e_{i}\right)}^{2}+E_{i},$$
where e is the reaction coordinate with the local equilibrium position $e_{i}$, M is the effective mass of the ‘defect molecule’ [47], $\omega_{i}$ is the vibrational frequency of mode i and $E_{i}$ represent the potential energy of states.

In terms of the relationship of energy to reaction coordinates in Equation (2), one can obtain the relationship of a nonradiative multiphonon transition process, as shown in Figure 9a. In the regular operation of the transistor, photons are not available, which means the increase of energy should only rely on many phonons during the transition process. In Figure 9a, $E_{12}$ and $E_{21}$ are the electron capture and emission energy barriers, respectively. $\lambda_{12}$ and $\lambda_{21}$ are the reorganization energies that reflects the strength of electron-phonon coupling, respectively. When an electric field is present at the location of the defect, the total energy difference between two states changes from $E_{2}$–$E_{1}$ to $E_{2}^{\prime }$–$E_{1}^{\prime }$ by $qV_{ox}$. For defects in the gate dielectric Al_2_O_3_, $V_{ox}$ is the potential difference at the defect from the interface. For defects in IGZO channel, $V_{ox}$ is the surface potential at the location of defects. Besides, the capture energy barrier and emission energy barrier also change to $E_{12}^{\prime }$ and $E_{21}^{\prime }$, as shown in Figure 9b. Then, the rates of transition between two states can be given [44],
(3)$$\{\begin{array}{c} k_{12}=N_{c}v_{th}\sigma e^{-E_{12}/kT}\\ k_{21}=nv_{th}\sigma e^{-E_{21}/kT} \end{array},$$
here, $N_{c}$ is the effective DOS in CBM, n is the electron density of active channel, $v_{th}$ is the thermal velocity of the electron, σ is the capture cross-section, k is the Boltzmann constant, and T is the absolute temperature. According to Equation (3), the effective capture/emission times and the probability of two states occupying at the moment with the infinity of t are given,
(4)$$\{\begin{array}{c} \tau_{c}=\frac{1}{k_{12}}=\tau_{0}e^{E_{12}/kT}\\ \tau_{e}=\frac{1}{k_{21}}=\frac{N_{c}}{n}\tau_{0}e^{E_{21}/kT}\\ P_{2}=\frac{k_{12}}{k_{12}+k_{21}}=\frac{1}{1+\frac{n}{N_{c}}e^{\frac{E_{2}-E_{1}}{kT}}}\\ P_{1}=\frac{k_{21}}{k_{12}+k_{21}}=\frac{1}{1+\frac{N_{c}}{n}e^{\frac{E_{1}-E_{2}}{kT}}} \end{array},$$
here, effective time constant $\tau_{0}$ incorporates the $N_{c}$, $v_{th}$ and σ, and is weakly dependent on bias and temperature [48].

Then, the relationship of total energy to coordinates is estimated by using a DFT method. The energy expression of charged state $V_{1}$ and charged state $V_{2}$ can be expressed as,
(5)$$\{\begin{array}{c} E_{V_{1}}\left(e\right)=\frac{1}{2}M\omega_{1}^{2}{\left(e-e_{1}\right)}^{2}\;\;\;\;\;\;\;\;\;\;\\ E_{V_{2}}\left(e\right)=\frac{1}{2}M\omega_{2}^{2}{\left(e-e_{2}\right)}^{2}+E_{s}\\ E_{V_{2}^{\prime }}\left(e\right)=E_{V_{2}}\left(e\right)-qV_{ox}\;\;\;\;\;\;\;\;\;\;\;\\ \;\;\;\;E_{s}=E_{V_{2}}\left(e_{2}\right)-E_{V_{1}}\left(e_{1}\right) \end{array},$$
here, $E_{V_{2}}\left(e_{1}\right)$ and $E_{V_{1}}\left(e_{2}\right)$ can be also calculated to define the quadratic function. The coefficient $\frac{1}{2}M\omega_{i}^{2}$ can be extracted from the parabolic potential energies.

#### 3.3.2. Charge Transition Process in Al_2_O_3_

By analyzing the nonradiative multiphonon process, one can discuss the kinetics of the charge transition process. Next, the interstitial hydrogen defect in Al_2_O_3_ gate dielectric will be discussed, which is a possible electron capture center under PBS in terms of the calculation of the transition levels. Since the capture of electrons by defects has a certain time constant, the probability of a defect capturing two electrons at the same time is very low. Firstly, the charge capture processes from $H_{i}^{+}$ to $H_{i}^{0}$ and $H_{i}^{0}$ to $H_{i}^{-}$ will be analyzed, respectively.

Figure 10 shows the total energy as a function of the reaction coordinates for the hydrogen interstitial in the Al_2_O_3_ dielectric layer and the energy barrier of the transition process. As can be seen in Figure 10a,b that for the transition process from $H_{i}^{+}$ to $H_{i}^{0},$ the reorganization energy $\lambda_{21}$ required is very high based on the method in Figure 9, reaching to 3.42 eV. The high reorganization energy indicates that the system has a strong electron-phonon coupling in this capture process. Otherwise, combined with the DOS’s results in Figure 6, it is found that the vibrational mode of the hydrogen-oxygen bond contributes to this electron capture process. Since the $H_{i}^{0}$ state has a higher equilibrium energy, the capture barrier will become relatively high, reaching to ~1.8 eV. According to Equation (5), this barrier will result in a low capture rate $k_{12}$. As $V_{ox}$ increases from 0V to 2V, the capture barrier decreases linearly from 1.8 eV to 0.3 eV. Then the capture rate will increase exponentially. For the transition process from $H_{i}^{0}$ to $H_{i}^{-}$ in Figure 10c,d that, the reorganization energy $\lambda_{21}$ is obviously different from that from $H_{i}^{+}$ to $H_{i}^{0}$. The different reorganization energy indicates that the vibrational modes contribute differently to the electronic transition process in the two processes. According to the DOS’s results, one can find that the vibrational mode of the adjacent aluminum ion around the hydrogen interstitial contributes to the $H_{i}^{0}$ to $H_{i}^{-}$ process, which is different for the process from $H_{i}^{+}$ to $H_{i}^{0}$. At the same time, although the equilibrium energy $E_{H_{i}^{-}}-E_{H_{i}^{0}}$ decreases continuously, while the probability of defects in $H_{i}^{-}$ state increases significantly, according to Equation (5). The capture barrier is basically equal to 0 eV, which means that the electron capture time $\tau_{c}$ from $H_{i}^{0}$ to $H_{i}^{-}$ is basically equal to $\tau_{0}$.

Overall, for the $H_{i}^{+}$ defect in the Al_2_O_3_ gate dielectric, when $V_{\mathrm{ox}}$ is greater than 0.52 V, the Fermi level is higher than the transition level ε (+1/−1), and the equilibrium energy of $H_{i}^{-}$ is lower than $H_{i}^{+}$, which means that the probability of defects in the state of −1 is higher than that in the state of +1. The electron capture of defects is a favorable energy process. However, the capture barrier is very high (about 1.4 eV), which makes the capture rate of $H_{i}^{+}$ to be very slow. After the electron capture is happened and $H_{i}^{0}$ is formed, the defect will generate electron capture again quickly by a capture time $\tau_{0}$ and forms $H_{i}^{-}$. As $V_{\mathrm{ox}}$ is continuously increased, the capture time $\tau_{c}$ from $H_{i}^{+}$ to $H_{i}^{0}$ will decline exponentially. These kinetic properties suggest that the electron capture process of hydrogen interstitial in Al_2_O_3_ is relatively slow at low voltage, but rapidly increases at high voltage, and eventually form the negative charge center in the gate dielectric oxide.

Next, we will discuss the transition process of oxygen vacancjes in the Al_2_O_3_ gate dielectric, which is a possible electron emission center under NBS based on the transition levels mentioned above. Similarly, we analyzed the emission process from $V_{O}^{0}$ to $V_{O}^{+}$ and from $V_{O}^{+}$ to $V_{O}^{2+}$, respectively. Figure 11 shows the total energy as a function of the reaction coordinate for the oxygen vacancies in the Al_2_O_3_ dielectric layer and the energy barrier of the transition process. As can be seen in Figure 11a,c that, for the two transition processes, the reorganization energies are relatively close, suggesting that the vibrational modes contribute to the electronic transition process, which may be the same in both processes. Combining with the DOS’s results, one can obtain that the vibrational modes of the adjacent aluminum ion around oxygen vacancies contribute to the electron emission process. According to Figure 11b, for the transition process from $V_{O}^{0}$ to $V_{O}^{+}$, the emission barrier decreases from 1.0 eV to 0 eV as $V_{\mathrm{ox}}$ increases from 0 V to 2 V. In Figure 11d, for the transition process from $V_{O}^{+}$ to $V_{O}^{2+}$, the emission barrier decreases from 0.5 eV to 0 eV as $V_{\mathrm{ox}}$ increases. And as $V_{\mathrm{ox}}$ is greater than 1.4 V, the emission barrier increases slightly.

Overall, for $V_{O}^{0}$ defect in the Al_2_O_3_ gate dielectric, when $\left|V_{\mathrm{ox}}\right|$ is greater than 0.69 V, the Fermi level is below the transition level ε (+2/−0). Thus, the defects undergo electron emission as an energy favorable process. At this point the emission barrier for the $V_{O}^{0}$ to $V_{O}^{+}$ is about 0.5 eV. Subsequently, $V_{O}^{+}$ then emits an electron again with a much lower emission barrier of about 0.2 eV and forms $V_{O}^{2+}$, which represents a significantly lower emission time $\tau_{e}$ for the process from $V_{O}^{+}$ to $V_{O}^{2+}$. As $\left|V_{ox}\right|$ increases further, the emission time $\tau_{e}$ for the transition process from $V_{O}^{0}$ to $V_{O}^{+}$ decreases rapidly due to a further decrease in the emission barrier, while the emission time $\tau_{e}$ for the transition process from $V_{O}^{+}$ to $V_{O}^{2+}$ gradually converges to $\tau_{0}$ and begins to increase as $V_{ox}$ is greater than 1.4 V. As a result, these kinetic properties suggest that the rate of electron emission process from the oxygen vacancies in Al_2_O_3_ increases as the voltage increases, and finally forms the positive charge center in the gate dielectric oxide.

#### 3.3.3. Charge Transition Process in CAAC-IGZO

Finally, we will discuss the electron emission process of the oxygen vacancies in the CAAC-IGZO active layer. Figure 12 shows the total energy as a function of the reaction coordinate for the oxygen vacancies in the CAAC-IGZO active layer and the energy barrier of the transition process, respectively. Based on the results mentioned above, four kinds of oxygen vacancies in CAAC-IGZO are similar. Here, we only discuss the transition process of $V_{O2}$. As can be seen in Figure 12a,c that, for the electron emission processes from $V_{O}^{0}$ to $V_{O}^{+}$ and from $V_{O}^{+}$ to $V_{O}^{2+}$, the reorganization energies are very close. The results imply that the vibrational modes contribute to the electronic emission process, which may be the same in both processes. The DOS’s results also display that the vibrational modes of the adjacent metal ion around oxygen vacancy contribute to the electron emission process. The lower reorganization energy represents a weaker electron-phonon coupling of the oxygen vacancies in the CAAC-IGZO active layer, as compared with the defect transition process in Al_2_O_3_. In Figure 12b, one can see that for the transition process from $V_{O}^{0}$ to $V_{O}^{+}$, the emission barrier decreases rapidly from 1.0 eV to 0 eV as $\left|V_{ox}\right|$ increases slightly at $\left|V_{ox}\right|$ being greater than 1.6 V. In Figure 12d, for the transition process from $V_{O}^{+}$ to $V_{O}^{2+}$, the emission barrier decreases from 0.6 eV to 0 eV as $\left|V_{ox}\right|$ increases and begins to increase when $\left|V_{ox}\right|$ is greater than 1.2 V. The emission barrier is approximately 0.2 eV, as $\left|V_{ox}\right|$ is equal to 2 V.

Overall, for the $V_{O}^{0}$ defect in the CAAC-IGZO active layer, when $V_{\mathrm{ox}}$ is greater than 0.73 V. the Fermi level is below the transition energy level ε (+2/−0), the defects undergo the electron emission as a favorable energy process. At that point, the emission barrier for the transition process from $V_{O}^{0}$ to $V_{O}^{+}$ is about 0.25 eV. Subsequently, the $V_{O}^{+}$ defect is then emitted again with a much lower emission energy of about 0.1 eV. As $\left|V_{ox}\right|$ increases further, the emission time $\tau_{e}$ for the transition process from $V_{O}^{0}$ to $V_{O}^{+}$ decreases rapidly to $\tau_{0}$ and begins to increase at $\left|V_{ox}\right|$ greater than 1.6 V, while the emission time $\tau_{e}$ for the transition process from $V_{O}^{+}$ to $V_{O}^{2+}$ converges to $\tau_{0}$ and begins to increase at $\left|V_{ox}\right|$ greater than 1.2 V. These kinetics properties suggest that the rate of the electron emission process increases with the increasing voltage, then begins to slow down when the voltage reaches about 1.6 V, and finally forming a positive charge center in the active layer.

### 3.4. Instability Effects in IGZO Device Induced by Charge Capture/emission Process

As mentioned above, the instability of the CAAC-IGZO FETs is mainly reflected in the drift of the threshold voltage under NBS and PBS. To uncover the instability, we then discuss the instability effects in IGZO devices induced by the charge capture/emission process. General speaking, the change of defect charge state caused by the charge capture/emission process will leave a new positive or negative charge center in oxide semiconductors and gate dielectric, which can be observed macroscopically as a change of fixed oxide charge per area ($Q_{ox}$) and affects the electrical performance of the device. The relationship of the threshold voltage drifts with $Q_{ox}$ is described as [44],
(6)$$\Delta V_{th}\left(V_{G}\right)=-\frac{\Delta Q_{ox}\left(V_{G}\right)}{C_{ox}},$$
here, $C_{ox}$ is the capacitance per area of gate dielectric. Universally, the capture/emission process of the oxygen vacancies does not quickly follow the gate bias change, which will be experimental recorded as a BTI effects. Under PBS, the electron capture process of hydrogen interstitial in the Al_2_O_3_ dielectric layer will decrease the value of $Q_{ox}$, and then induce a positive drift of the threshold voltage. In a 2T0C DRAM cell, this PBS-induced positive drift occurs within the write transistor and read transistor of the write operation and results in lower open-state current and slower write operations. In addition, the large capture barrier and capture time of the process from $H_{i}^{+}$ to $H_{i}^{0}$ will cause a small drift of the threshold voltage at lower temperatures and low positive gate bias voltage. The experiment displays that the Al_2_O_3_ gate dielectric with 1.1% hydrogen concentration may result in a positive but small $\Delta V_{th}$ at 25 °C under PBS [33]. While the trap level *E_t_* tested by IPE measurements is 0.69 eV above the Fermi level, which is similar to our results. However, the $\Delta V_{th}$ will become negative as the temperature raising. The reason is that under higher stress, trapped hydrogen in the gate-dielectric is released, diffuses to IGZO layer. To overcome the instability, some approaches can be taken to control the hydrogen concentration in Al_2_O_3_ dielectric layer, such as replacing water with ozone as a precursor for ALD of the gate dielectric.

Under NBS, the electron emission process of the oxygen vacancies in the Al_2_O_3_ dielectric layer and CAAC-IGZO active layer will lead to the increase of $Q_{ox}$, and then induce a negative drift of the threshold voltage. In a 2T0C DRAM cell, the NBS-induced negative drift occurs within the write transistor of the read operation and results in larger off-state current and shorter retention time. Generally, the transistor is in off-state under a negative gate bias. The phenomenon of the negative drift of the threshold voltage will be observed when transistor is in on-state again and one can find a hysteresis of the current-voltage curve. To better understand the phenomenon of the negative drift of the threshold voltage induced, one firstly need to distinguish between the different contributions of the oxygen vacancies in the Al_2_O_3_ dielectric layer and CAAC-IGZO active layer to the threshold voltage drift, respectively. According to the calculations, the behavior of the oxygen vacancies in the Al_2_O_3_ dielectric layer and CAAC-IGZO active layer is essentially similar. However, it should be noted, that $V_{ox}$ is the surface potential at the defect for the active layer and the potential difference at the defect for the dielectric layer. It is well known that when the gate bias of transistor is below the threshold voltage, more voltage will fall on the active layer thus changing the surface potential and the carrier concentration. Therefore, the oxygen vacancies in CAAC-IGZO active layer may contribute more to the negative drift of the threshold. The effect of the oxygen vacancies under NBS is very similar to that in amorphous IGZO [31], except that the oxygen vacancies in CAAC-IGZO do not lead to the formation of shallow donors. To overcome the instability, some approaches can be taken to control the concentration of the oxygen vacancy, such as annealing in an oxygen atmosphere.

## 4. Conclusions

In this work, the charge capture and emission properties of different defects in the CAAC-IGZO FETs have been studied by using the first-principles calculations. The results display that the hydrogen interstitials in the Al_2_O_3_ dielectric layer are probable electron emission center, as well as the oxygen vacancies in Al_2_O_3_ dielectric layer and CAAC-IGZO active layer as the electron emission center. More importantly, by combining the nonradiative multiphonon theory, we discussed the configuration change and kinetics of different transition processes in detail. It is found that a high capture barrier about 1.4 eV for electron capture process of hydrogen interstitial in Al_2_O_3_ dielectric layer and an emission barrier of about 0.5 eV and 0.25 eV for the electron emission process of the oxygen vacancies in Al_2_O_3_ dielectric layer and CAAC-IGZO active layer, can be formed, respectively. The formation of these charge centers will seriously affect the charge capture/emission process and then induce the instability of the CAAC-IGZO FETs, where hydrogen interstitials cause a positive drift of the threshold voltage and oxygen vacancies cause a negative drift of the threshold voltage. This work helps understand the mechanisms of instability of CAAC-IGZO transistor caused by carrier capture/emission process and provides a theoretical reference for the improvement of the experiment.

## Figures and Tables

**Figure 1 materials-16-02282-f001:**
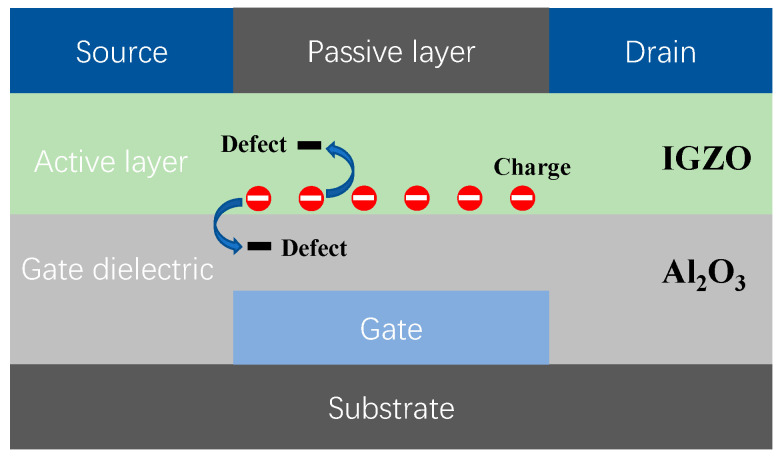
Illustration of carrier trapping process in CAAC-IGZO FET.

**Figure 2 materials-16-02282-f002:**
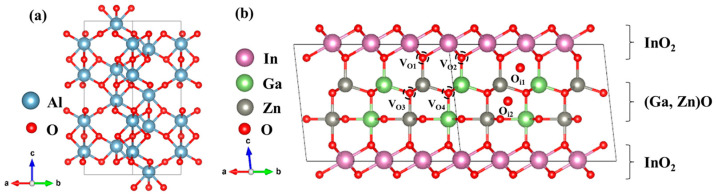
Crystal structure of (**a**) α-Al_2_O_3_ and (**b**) CAAC-IGZO. The structural model is drawn with VESTA [36].

**Figure 3 materials-16-02282-f003:**
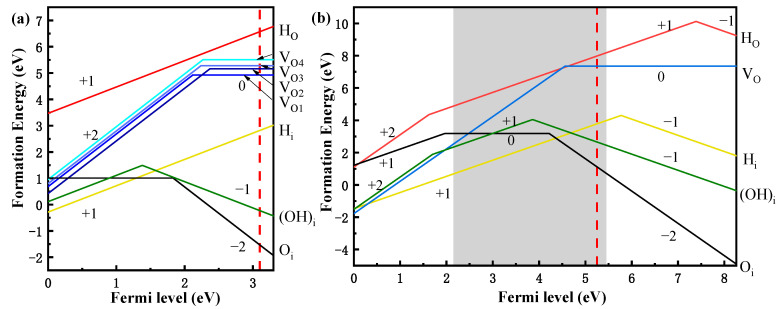
The formation energy of defects under O-rich condition for CAAC-IGZO (**a**) and Al_2_O_3_ (**b**). The red dotted line represents the Fermi level in the absence of an electric field and the grey area in (**b**) represents the area of band gap in CAAC-IGZO.

**Figure 4 materials-16-02282-f004:**
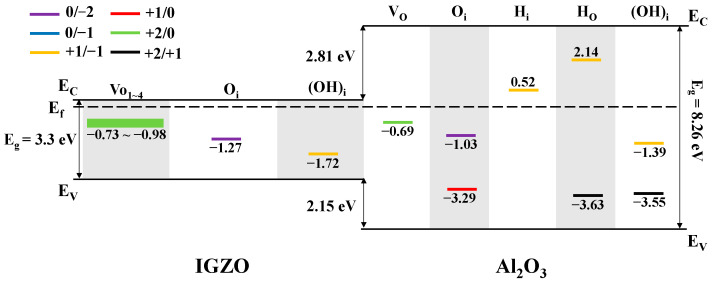
The band alignment and charged transition level of defects in Al_2_O_3_/IGZO. The numbers in the figure represent the energy from the transition level to the Fermi level of CAAC-IGZO.

**Figure 5 materials-16-02282-f005:**
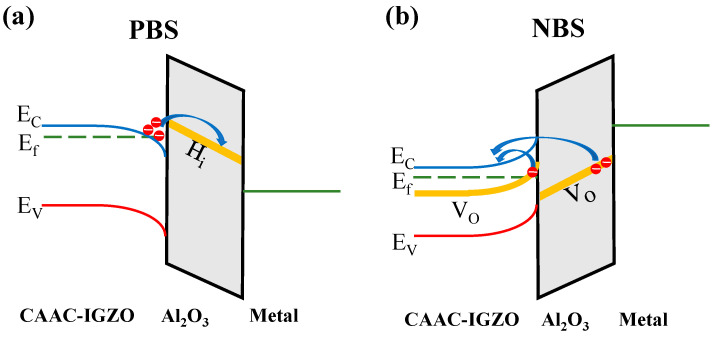
The schematics diagram of electron trapping process under PBS (**a**), and the electron emission process under NBS (**b**). Electrons are exchanged between the intrinsic defects and the conduction band of the CAAC-IGZO. Under PBS, electrons will be trapped in H_i_ defects in Al_2_O_3_. Under NBS, electrons will be emitted from Vo defect in CAAC-IGZO and Al_2_O_3_.

**Figure 6 materials-16-02282-f006:**
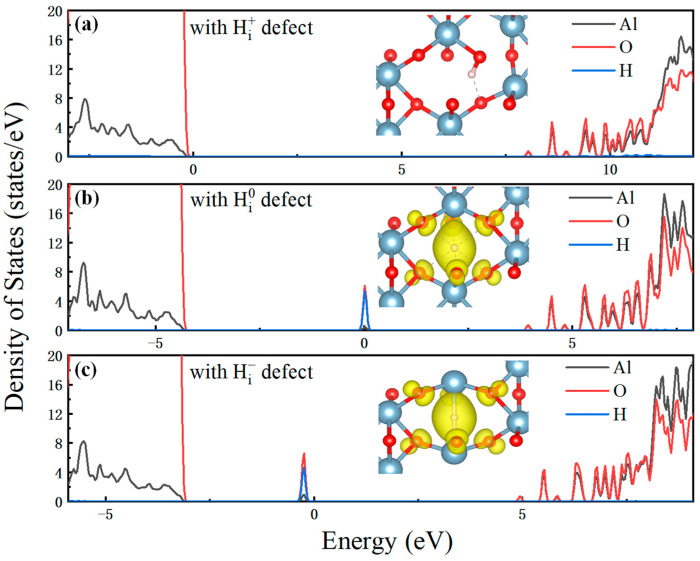
DOS of hydrogen interstitial with the charge states of (**a**) +1, (**b**) 0, and (**c**) −1 in Al_2_O_3_ dielectric layer, respectively. In inset, the occupied state orbitals are marked by yellow isosurface.

**Figure 7 materials-16-02282-f007:**
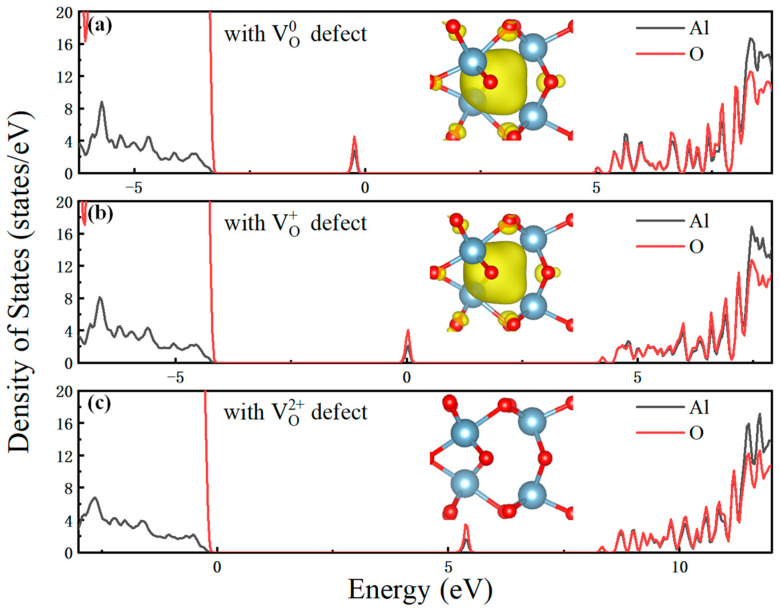
DOS of oxygen vacancy with the charge states of (**a**) 0, (**b**) +1, and (**c**) +2 in Al_2_O_3_ dielectric layer, respectively. In the inset, the occupied state orbitals are marked by yellow isosurface.

**Figure 8 materials-16-02282-f008:**
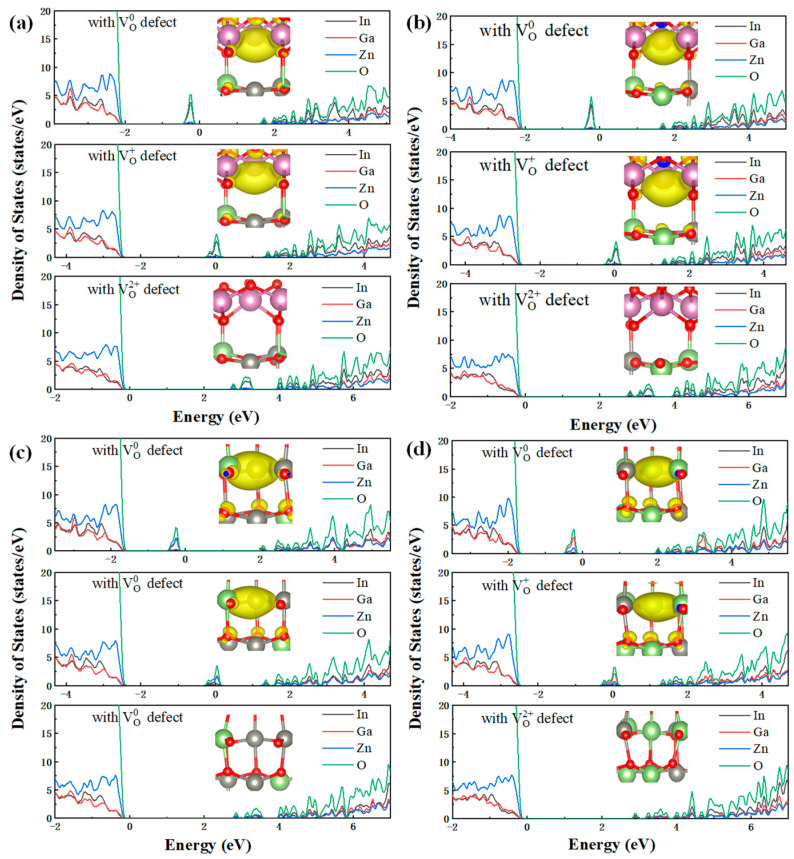
DOS of oxygen vacancy (**a**) $V_{O1}$, (**b**) $V_{O2}$, (**c**) $V_{O3}$ and (**d**) $V_{O4}$ with different charge states in CAAC-IGZO active layer.

**Figure 9 materials-16-02282-f009:**
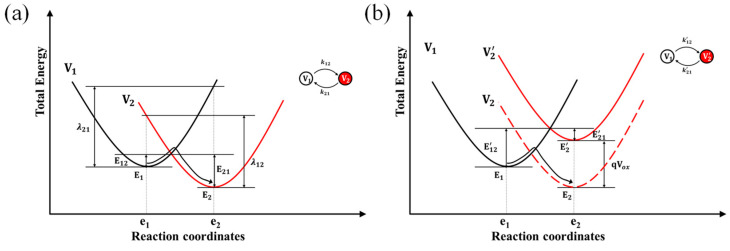
Relationship of total energy to reaction coordinate of nonradiative multiphonon transition process without electric field (**a**) and with an electric field (**b**). $V_{1}$ and $V_{2}$ represent the neutral state and negative charge state of the defect, respectively. $V_{2}^{\prime }$ represents the negative charge state of defect when an electric field is present. $E_{12}$ is the electron capture energy barrier, and $E_{21}$ is the electron emission energy barrier. The vibrational energy of the system is modeled by $E_{12}$ and $E_{21}$.

**Figure 10 materials-16-02282-f010:**
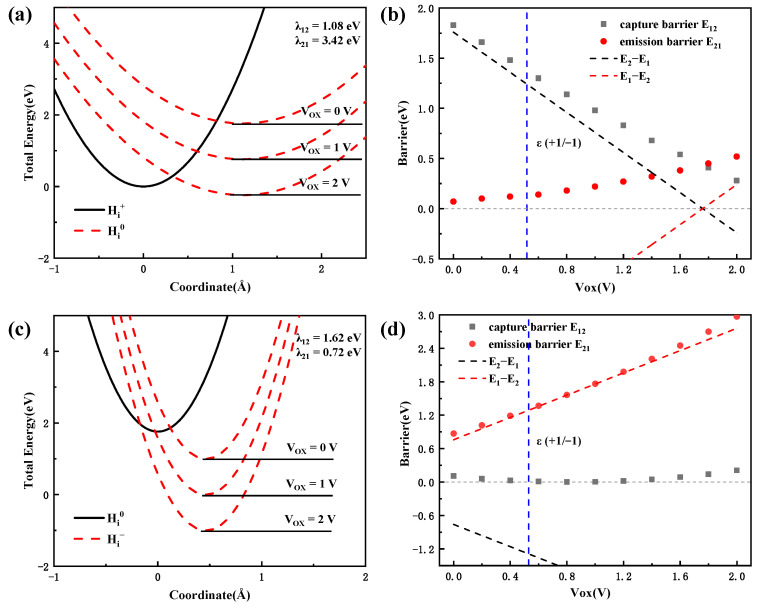
The total energy as a function of the reaction coordinate for the hydrogen interstitial in Al_2_O_3_ dielectric layer, (**a**) from $H_{i}^{+}$ to $H_{i}^{0}$ and (**c**) from $H_{i}^{0}$ to $H_{i}^{-}$. The energy barrier as a function of $V_{\mathrm{ox}}$, (**b**) from $H_{i}^{+}$ to $H_{i}^{0}$ and (**d**) from $H_{i}^{0}$ to $H_{i}^{-}$ process. The red dashed line is the total energy curve for the charge state at different $V_{\mathrm{ox}}$. The blue dash line represent the transition level ε (+1/−1).

**Figure 11 materials-16-02282-f011:**
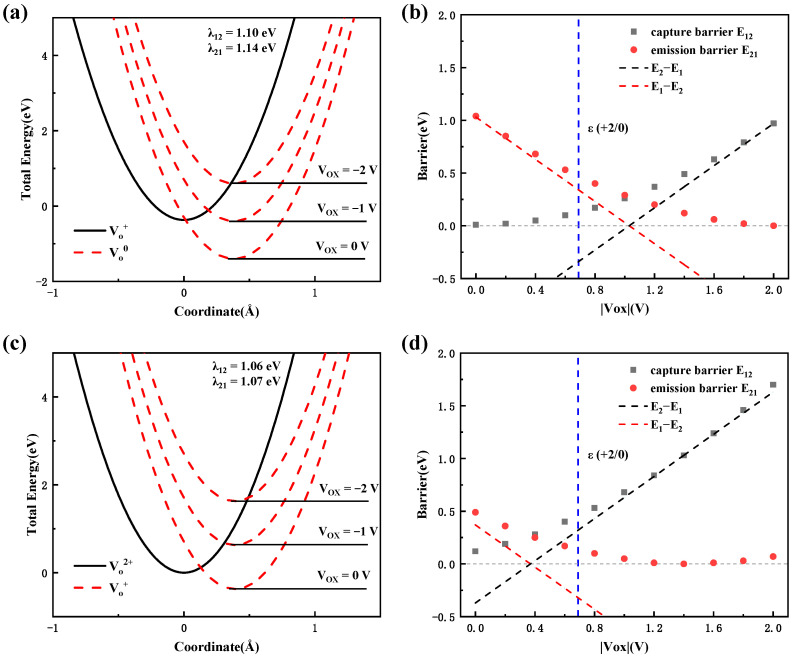
The total energy as a function of reaction coordinates for the oxygen vacancies in Al_2_O_3_ dielectric layer, (**a**) from $V_{O}^{0}$ to $V_{O}^{+}$ and (**c**) from $V_{O}^{+}$ to $V_{O}^{2+}$. The red dashed line is the total energy curve for the charge state at different $V_{\mathrm{ox}}$. The energy barrier as a function of $V_{\mathrm{ox}}$, (**b**) from $V_{O}^{0}$ to $V_{O}^{+}$ and (**d**) from $V_{O}^{+}$ to $V_{O}^{2+}$. The blue dash line represent the transition level ε (+2/0).

**Figure 12 materials-16-02282-f012:**
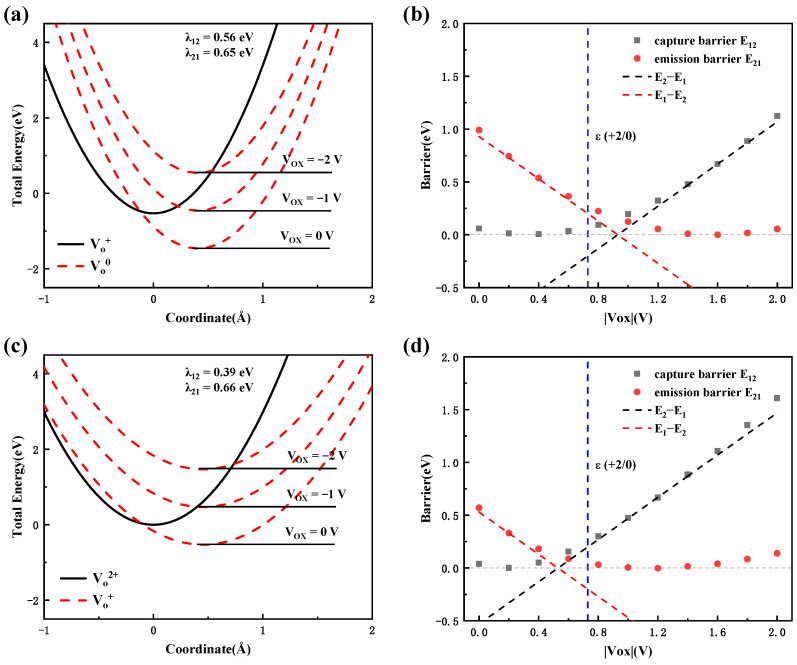
The total energy as a function of reaction coordinates for the oxygen vacancies in CAAC-IGZO active layer, (**a**) from $V_{O}^{0}$ to $V_{O}^{+}$ and (**c**) from $V_{O}^{+}$ to $V_{O}^{2+}$. The red dashed line is the total energy curve for the charge state at different $V_{\mathrm{ox}}$. The energy barrier as a function of $V_{\mathrm{ox}}$, (**b**) from $V_{O}^{0}$ to $V_{O}^{+}$ and (**d**) from $V_{O}^{+}$ to $V_{O}^{2+}$. The blue dash line represent the transition level ε (+2/0).

## Data Availability

All data used in this work will be available upon reasonable request.

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
