# Peer review of "Charge Trapping and Emission Properties in CAAC-IGZO Transistor: A First-Principles Calculations"

_materials, 2023, doi:10.3390/ma16062282_

Round 1
Reviewer 1 Report
In this manuscript, the authors described the studies of the charge capture and emission properties of different defects in CAAC-IGZO transistor using the first-principles calculations. They discussed several properties regarding charge transitions. The manuscript has merit, but some questions could be discussed in detail. The questions are following:
In the abstract, the acronyms, such as CAAC-IGZO FET and DRAM, should be defined.
Could the authors contextualize the DRAM applications once it’s the focus of the devices studied in the manuscript?
The transition levels ε(0/-2) of Oi in the CAAC-IGZO and in Al2O3 are close to the transition levels ε(+2/0) of Vo. Under NBS, what would be the effects of this? Would a competition occur? What would be the consequences?
How significantly the change in the activation barrier of the charge capture/emission process influence the kinetics of the charge capture/emission process?
Is the charge capture/emission process occurring in the superficial layer of the materials or in the bulk? For a bulk effect, should not be considered the volumetric capacitance (regarding equation 6) for calculation of the threshold voltage?
Author Response
Thank you for your kindly review. Please see the attachment.

Reviewer 2 Report
This study is interesting for publishing, it contains valuable data
- May you mention what is CAAC-IGZO in the abstract
- Can you make the abstract more attractive by inserting some of your real results and findings into it
- In the computational method, do you think that The cut-off energy for the wave function of less than 500 ev can change the calculations? And why
- Can you compare your calculations of measuring and controlling the defects such as oxygen vacancies with Kroger-Vink notation for oxygen vacancies calculations
- Role of oxygen vacancies in vanadium oxide and oxygen functional groups in graphene oxide for room temperature CO2 gas sensors - ScienceDirect
- In Fig 5, which one is having a higher Defect drafting potential
- May you mention the relation between the DOS of hydrogen interstitial and oxygen capacity
- Can you please check your results with the recent literature results and make a small comparison based on your results? I see that these results are so interesting
Author Response
Thanks for your kindly review. Please see the attachment.

Round 2
Reviewer 2 Report
Thank you for your efforts in answering the required questions.